# Responses of Crop Yield, Soil Fertility, and Heavy Metals to Spent Mushroom Residues Application

**DOI:** 10.3390/plants13050663

**Published:** 2024-02-28

**Authors:** Qichao Tang, Weijia Liu, Han Huang, Zhaohui Peng, Liangji Deng

**Affiliations:** 1College of Resources, Sichuan Agricultural University, Chengdu 611130, China; tangqichao0125@163.com (Q.T.); liuweijia27@163.com (W.L.); 2Institute of Agricultural Bioenvironment and Energy, Chengdu Academy of Agriculture and Forestry Sciences, Chengdu 611130, China; pengp0780@163.com; 3College of Economics and Management, Xinjiang Agricultural University, Urumqi 830052, China; 18328717493@163.com

**Keywords:** mushroom residue, crop yield, soil fertility, heavy metal, agro-ecosystems

## Abstract

Waste mushroom residues are often returned to fields as organic amendments. Here, we estimated the effects of the continuous applications of different spent mushroom substrates for 2 years on crop yields, soil nutrients, and heavy metals in paddy fields. The study comprised seven treatments: no fertilization (CK) and mineral NPK fertilizer (CF), as well as NPK fertilizer combined with Enoki mushroom residue (EMR50), Oyster mushroom residue (OMR50), *Auricularia polytricha* mushroom residue (APR50), Shiitake mushroom residue (SMR50), and *Agaricus bisporus* residue (ABR50). The grain yield was highest under the APR50 treatment. The short-term application of waste mushroom residue significantly increased SOC, TN, TP, and TK content relative to the CK treatment. The SOC, TP, and TK were highest under ABR50. Both total Cr and Cd contents were highest under CF treatment. The highest cumulative ecological risk was observed under OMR50 treatment. In addition, crop yield was positively correlated with SOC, TN, TP, and TP. Our results highlight that further research and innovation are needed to optimize the benefits and overcome the challenges of mushroom residue application.

## 1. Introduction

The Chengdu Plain is the main production region for wheat and rice in China, earning the moniker “TianFu Granary” due to its significant role in food production [1,2,3]. In the past few decades, grain production in the Chengdu Plain has increased significantly, mainly due to the investment in synthetic fertilizers [4,5,6]. Excessive and prolonged use of synthetic fertilizers, particularly nitrogen-based ones, can lead to various environmental issues, such as soil acidification, water pollution, and nitrous oxide emissions [7,8,9,10]. To address these concerns, China’s Ministry of Agriculture and Rural Affairs introduced the “zero growth of chemical fertilizers by 2020” policy [11,12,13]. Government initiatives highlight the importance of enhancing fertilizer management practices in crop production.

Waste mushroom residue emerges as a byproduct within the context of mushroom production [14,15]. Mushroom cultivation frequently leads to the production of a substantial amount of waste mushroom residue. This is mainly attributed to the relatively low utilization rate of the initial substrate, such as crop straw, cottonseed shell, and sawdust, which is currently only at 40.0%. This inefficiency leads to the production of approximately 5.0 kg of mushroom residue for every 1.0 kg of mushrooms cultivated [14]. Unfortunately, waste mushroom residue is frequently disposed of through unregulated piling or burning practices, resulting in the wastage of valuable lignocellulosic resources and occupying significant land areas [15]. Furthermore, these practices contribute to severe environmental issues such as mosquito breeding, the release of odorous gases, and groundwater pollution [16,17]. Sichuan is a major province in the production and consumption of mushroom in China. Statistics show that the production of waste mushroom residue in Sichuan Province reached 2.3 × 10^8^ tons in 2020.

Returning waste mushroom residues to fields is considered an environmentally friendly agronomic practice that is suitable for reducing agricultural pollution. Meanwhile, waste mushroom residue is rich in macro- and micronutrients, making it a viable option for organic fertilizer [18,19]. Previous studies have found that the appropriate combination of mineral fertilizers and waste mushroom residues is beneficial to improving and maintaining soil quality and plant growth [20,21,22]. As an important producer of mushrooms in China, Sichuan mainly produces Enoki mushroom, Oyster mushroom, *Auricularia polytricha* mushroom, Shiitake mushroom, and *Agaricus bisporus* [23]. Yet, the composition of different waste mushroom residues varies depending on the base substrate and its supplementation. Therefore, the effects of different types of waste mushroom residues application on crop yield and soil fertility still require further research.

Assessing the potential risk of heavy metal accumulation in soils is critical for human health, especially considering the promotion of waste mushroom residues as a vital source of nutrients for cropland [24,25,26]. Waste mushroom residues often contain non-essential, non-degradable heavy metals that have the potential to accumulate in soil when mushroom residues are applied [27]. Previous studies have shown that the prolonged utilization of mushroom residues can result in an increase in soil heavy metal content [23,28,29,30]. Simultaneously, some studies have observed the accumulation of heavy metal in soil due to the prolonged use of synthetic fertilizers [31,32,33,34]. The pollution status of cropland is an indicator of human health as it has a significant impact on food quality. Therefore, it is important to estimate the heavy metal pollution status of agricultural land.

In this study, we evaluated the effects of the short-term application of different waste mushroom residues on rice–wheat rotation soil in the Chengdu Plain. The aims of our study were to estimate how crop yield, soil fertility, and heavy metal contents are influenced by waste mushroom residue amendment. 

## 2. Results

### 2.1. Crop Yield and Soil Fertility

Short-term fertilization significantly affected crop yield and soil nutrient contents (*p* < 0.05) (Table 1 and Figure 1). The joint application of mushroom residue and NPK fertilizer resulted in a significant enhancement of crop yield, as well as increases in soil organic carbon (SOC), total nitrogen (TN), and total phosphorus (TP) compared to the treatment with CF treatment. In comparison to the CK treatment, the increases in grain yield under CF, EMR50, OMR50, APR50, SMR50, and ABR50 were by 49.1%, 66.1%, 50.0%, 84.4%, 82.2%, and 78.0%, respectively. Compared to the CK treatment, the increases in SOC under CF, EMR50, OMR50, APR50, SMR50, and ABR50 were by 4.7%, 17.0%, 27.1%, 24.0%, 21.1%, and 31.0%, respectively. The CF, EMR50, OMR50, APR50, SMR50, and ABR50 treatments increased TN content by 34.0%, 99.0%, 112.0%, 101.5%, 111.3%, and 88.7%, respectively, TP content was increased by 107.2%, 190.8%, 164.4%, 146.7%, 171.0%, and 232.2%, respectively, and TK content was increased by 50.6%, 54.2%, 45.2%, 46.4%, 56.9%, and 73.8%, respectively, relative to the CK treatment. 

### 2.2. Heavy Metals

Short-term fertilization significantly affected the heavy metal accumulation in soil (Figure 2). Compared to CK treatment, CF, EMR50, OMR50, APR50, SMR50, and ABR50 treatments increased total Cr content by 31.4%, 28.4%, 28.3%, 21.1%, 30.2%, and 17.5%, respectively. Compared to CK treatment, CF, OMR50, SMR50, and ABR50 treatments increased total Cd content by 11.2%, 4.4%, 2.0%, and 11.2%, but EMR50 and APR50 treatments decreased total Cd content by −0.3% and −10.6%, respectively. The EMR50, OMR50, and APR50 treatments increased total Pb content by 13.2%, 17.0%, and 9.5%, and the CF, SMR50, and ABR50 treatments decreased total Pb content by −6.5%, −4.5%, and −1.2%, respectively, relative to CK treatment. Compared to the CK treatment, CF, EMR50, OMR50, APR50, SMR50, and ABR50 treatments increased the RI index by 23.7%, 23.1%, 24.1%, 16.0%, 21.7%, and 13.8%, respectively. 

### 2.3. Relationships between Soil Nutrients and Crop Yield and Heavy Metals

Crop yield was positively correlated with SOC (r = 0.61, *p* < 0.01), TN (r = 0.47, *p* < 0.05), and TP (r = 0.58, *p* < 0.01). SOC was positively correlated with TN (r = 0.60, *p* < 0.01), TP (r = 0.76, *p* < 0.001), and TK (r = 0.57, *p* < 0.01). Both Cr and Pb were significantly correlated to TN (r = 0.48, *p* < 0.05) (Figure 3).

## 3. Discussion

Globally, the practice of fertilization, particularly the use of organic fertilizers such as mushroom residue in field crops, has become widespread. This is attributed to the presence of essential macro- and micronutrients in mushroom residue. In China, with significant advancements in the mushroom industry, the application of mushroom residue to field crops as a fertilizer source has become a standard practice. Furthermore, at the national level in China, there is encouragement for the application of mushroom residue to agricultural land. Nevertheless, the prolonged use of waste mushroom residue may pose environmental threats owing to the presence of non-essential heavy metals in waste mushroom residue.

### 3.1. Response of Crop Yield and Soil Nutrients to Fertilization 

The results of this study showed that short-term fertilization with different treatments had significant effects on crop yield and soil nutrient contents. The combined application of mushroom residue and NPK fertilizer (EMR50, OMR50, APR50, SMR50, and ABR50) outperformed the sole application of NPK fertilizer (CF) in terms of increasing crop yield, SOC, TN, and TP. This indicates that mushroom residue can be used as an effective organic amendment to enhance the fertility and productivity of agricultural soils.

Meanwhile, the findings of this study showed that the application of waste mushroom residue significantly increased crop yield and soil physicochemical properties, compared to no fertilization. These results are in agreement with previous studies that reported the beneficial effects of mushroom residue application on crop performance and soil health. For instance, Medina et al. (2012) showed that waste mushroom residue application can enhance soil nutrients and enzyme activities [35]. Kwiatkowski et al. (2021) found that waste mushroom residue application can improve yield quality [36]. Ma et al. (2021) observed that the application of mushroom residue can improve soil nutrients, enhance leaf nutrients, and suppress weeds [37]. Paredes et al. (2016) found that the application of waste mushroom residue can improve soil fertility and crop yield [38]. Chen et al. (2022) reported that the substitution of chemical fertilizer with waste mushroom residue can increase soil nutrients, such as SOM, TN, AN, TP, AP, TK, and AK [23]. The increase in crop yield under the combined application of mushroom residue and NPK fertilizer could be attributed to several factors. First, waste mushroom residue serves as a valuable source of organic matter and essential nutrients, especially nitrogen, phosphorus, and potassium, crucial for plant growth and development [39]. Second, the inclusion of waste mushroom residue in soil can enhance its physical and chemical properties, including water holding capacity, pH, and electrical conductivity, thereby improving soil fertility and productivity [29,40]. Third, waste mushroom residue has the capability to stimulate microbial activity and diversity in soil. This stimulation promotes the decomposition of organic matter and nutrient release and acts as deterrent to soil-borne pathogens and pests [41]. The observed positive correlations between crop yield and SOC, TN, and TP suggest that these soil parameters are crucial indicators of soil quality and plant nutrition. The increases in SOC, TN, TP, and TK under the mushroom residue treatments also imply that mushroom residue can improve the soil carbon and nitrogen cycles and provide sufficient phosphorus and potassium for crop growth. 

However, there are some limitations and challenges in the application of mushroom residue as an organic fertilizer. First, the quality and composition of mushroom residue may vary depending on the mushroom species, substrate materials, cultivation conditions, and harvesting methods [42]. It is necessary to monitor and adjust the mushroom residue characteristics to ensure its suitability and safety for different crops and soils. Second, the optimal rate and frequency of mushroom residue application may depend on various factors, such as crop type, soil type, climate, and management practices [23]. Third, the transportation and storage of mushroom residue may pose some environmental and logistical problems, such as greenhouse gas emissions, odor, and contamination. Therefore, it is essential to develop and adopt efficient and sustainable technologies and strategies to process and utilize mushroom residue, such as vermicomposting, pelletizing, and biogas production [43].

### 3.2. Response of Heavy Metals to Organic and Chemical-Organic Fertilization 

Our results indicated that short-term fertilization with different treatments had significant effects on heavy metal accumulation in soil. The application of mushroom residue and NPK fertilizer enhanced the total Cr content, while the application of NPK fertilizer alone (CF) increased the total Cd content, compared to the no fertilization treatment (CK). The effects of mushroom residue and NPK fertilizer on the total Pb content varied depending on the type of mushroom residue. The application of mushroom residue and NPK fertilizer also increased the RI index, which is an indicator of the potential ecological risk of heavy metals in soil. These results indicate that mushroom residue can be a source of heavy metals in soil, and that its application should be carefully monitored and controlled to avoid excessive accumulation and environmental pollution.

Our results showed that mushroom residue application significantly increased the total contents of Cr, Cd, and Pb in soil, compared to no fertilization. This result is in line with previous studies that reported the accumulation of heavy metals in soil after mushroom residue application. Mushroom residue contains various heavy metals, such as Cr, Cd, and Pb, which are derived from the substrate materials, cultivation conditions, and the harvesting methods of mushroom production [44]. Moreover, mushroom residue can affect the availability and mobility of heavy metals in soil by altering the soil pH, organic matter, cation exchange capacity, and microbial activity [29]. Notably, mushroom residue can pose a risk of heavy metal contamination in the soil–plant system, which may affect food safety and human health [28]. Therefore, it is necessary to evaluate and regulate the quality and quantity of mushroom residue applied to the soil, and to adopt appropriate management practices to minimize the adverse effects of heavy metals. For example, it is important to select mushroom residue with low heavy metal content, to mix mushroom residue with other organic amendments, to apply mushroom residue at a suitable rate and frequency, and to monitor the heavy metal levels in the soil and crops.

Overall, this study provides empirical evidence for the beneficial use of waste mushroom residue in soil amendment to improve soil quality and enhance crop yield. Meanwhile, mushroom residue can be a source of heavy metals in soil, and its application can affect heavy metal accumulation and ecological risk in soil. Further research and innovation are needed to optimize the benefits and overcome the challenges of mushroom residue application [45,46,47].

## 4. Materials and Methods

### 4.1. Experiment Aera

A field experiment on a rice–wheat cropping system was established in 2020 at the Modern Agriculture Base of Sichuan Agricultural University, Qiquan village, Chongzhou (30°33′27″ N, 103°38′34″ E), in the west of the Chengdu Basin, China. The field experiment site has a subtropical humid monsoon climate, characterized by an average annual precipitation of 1012 mm and a temperature of 15.9 °C. The soil texture in the experimental field is clay loam. Some physio-chemical characteristics of the top soil depth (0–20 cm) at the start of the experimental were as follows: soil pH, 6.6; bulk density, 1.22 g cm^−3^; SOC (soil organic carbon content), 18.7 g kg^−1^; total nitrogen content, 2.0 g kg^−1^; total phosphorus content, 1.0 g kg^−1^; total potassium content, 8.7 g kg^−1^; total Pb content, 40.5 mg kg^−1^; total Cd content, 0.24 mg kg^−1^; total Cr content, 101.3 mg kg^−1^.

### 4.2. Experimental Design

The experiments were conducted using a completely randomized block design, with each fertilization treatment replicated three times. There was a 1.5 m wide buffer zone between plots, and the plot size was 5 × 4 m. Wheat row spacing was 20 cm, and rice row spacing was 40 cm. The following seven treatments were established: CK, no fertilizer; CF, chemical fertilizer comprising 330 kg ha^−1^ N, 165 kg ha^−1^ P_2_O_5_, 165 kg ha^−1^ K_2_O, representing the high-yielding fertilization regime used by the local smallholder farmer; EMR50, where the Enoki mushroom residues provided 50% N and chemical fertilizer provided 50% N; OMR50, where the Oyster mushroom residues provided 50% N and chemical fertilizer provided 50% N; APR50, where the *Auricularia polytricha* mushroom residues provided 50% N and chemical fertilizer provided 50% N; SMR50, where the Shiitake mushroom residues provided 50% N and chemical fertilizer provided 50% N; and ABR50, where the *Agaricus bisporus* residues provided 50% N and chemical fertilizer provided 50% N (Figure 4). The total amount of fertilizer N input in each mushroom residue application treatment was equal to the CF treatment. Further details regarding the basal information and application rates of different waste mushroom residues are provided in Table 2. The waste mushroom residue was sourced from edible mushroom cultivation institutions in the Chengdu Plain. For rice cultivation, 40% of the N fertilizer was applied as basal fertilizer, 40% at the tillering stage, and 20% at the booting stage. For wheat cultivation, 50% of the N fertilizer was applied as basal fertilizer, 25% at the seedling stage, and 25% at the jointing stage. P and K fertilizers and fresh waste mushroom residue were applied annually before plowing as basal fertilizers. All basal fertilizers and mushroom residue were evenly spread on the soil surface by hand and immediately plowed into the plow layer. The plowing depth was 20 cm, first using a plow and then a harrow. Fertilized plots and unfertilized plots were uniformly plowed. All agronomic management practices, expect fertilization, were consistent across all treatments. The wheat and rice were harvested at full maturity.

### 4.3. Soil Sampling and Analysis

The soil samples (in 0–20 cm) were randomly collected from seven different treatments in each plot following crop harvest. Composite soil samples were stored in sterilized ziplocked bags and transported to the laboratory. Plant roots and organic debris were removed prior to analysis. The fresh soil samples were gently fragmented along natural break points and sieved (<2 mm), while forceps were used to eliminate plant roots and organic debris. Before analysis, the soil was air dried, ground, and sieved to 0.25 mm. Soil organic carbon content was determined following the K_2_Cr_2_O_7_ oxidation method. Soil total nitrogen (N), phosphorus (P), and potassium (K) contents were determined following the Carter and Gregorich method [48]. For the extraction of the total cadmium (Cd), chromium (Cr), and lead (Pb) contents in soil, samples (<0.15 mm) were digested by using a mixture of HNO_3_–HCl (3:1). An atomic absorption spectrophotometer (FAASZ-5000, Hitachi, Tokyo, Japan) was utilized to analyze the concentrations of Cr, Cd, and Pb. The determination of heavy metals in waste mushroom residue involved using an extraction solution of concentrated acids HNO_3_ and HClO_4_ [32].

Potential ecological risk indicates the sensitivity of a biological community to pollutants and elucidates the level of ecological risk attributed to heavy metals. The potential ecological risk associated with the cumulative ecological risk (RI) was determined through the following estimation:$$RI=\sum_{i}^{n}T^{i}\times \frac{C_{x}^{i}}{C_{b}}$$
where $C_{x}^{i}$ represents the concentration of an individual heavy metal in sample x and $C_{b}^{i}$ is concentration of heavy metal in reference soil (national background value). The national background values for Cr, Cd, Pb, and Zn were 90 mg kg^−1^, 0.2 mg kg^−1^, 35 mg kg^−1^, and 10 mg kg^−1^, respectively [49]. $T^{i}$ denotes the biological toxicity of each metal (i); for instance, Ti values for Cd, Cr, Pb, and Zn were 30, 2, 5, and 1, respectively [50]. The sites were classified into four categories based on the RI value: RI < 150 indicating low ecological risk; 150 < RI < 300 indicating moderate ecological risk; 300 < RI < 600 indicating considerable ecological risk; and RI > 600 indicating very high ecological risk.

### 4.4. Data Analysis and Statistics

Statistical analyses were conducted using the SPSS 20.0 software package for Windows (SPSS Inc., Chicago, IL, USA). Prior to analysis of variance (ANOVA), data normality was assessed using the Kolmogorov–Smirnov test. In cases where normality needed improvement, the data were ln transformed. Significant differences among treatments were evaluated through one-way ANOVA, followed by the Tukey test for multiple comparisons at a significance level of *p* < 0.05.

## 5. Conclusions

Our study explored the impact of short-term fertilization with various treatments on crop yield, soil nutrient contents, and soil heavy metal accumulation. The findings revealed that the combined application of mushroom residue and NPK fertilizer led to a significant increase in crop yield and soil nutrient contents, compared to the sole application of NPK fertilizer or no fertilization. However, it was observed that the application of mushroom residue also enhanced the total contents and potential ecological risk of heavy metals in soil, particularly Cr and Cd. Our findings underscore the potential of mushroom residue as a valuable organic fertilizer to enhance soil fertility and productivity in agriculture. However, it emphasizes the importance of careful monitoring and control of mushroom residue application to prevent excessive accumulation and environmental pollution of heavy metals. This research contributes to the advancement of a circular economy in the mushroom sector and plays a role in safeguarding the soil environment amid global climate change and food security concerns.

## Figures and Tables

**Figure 1 plants-13-00663-f001:**
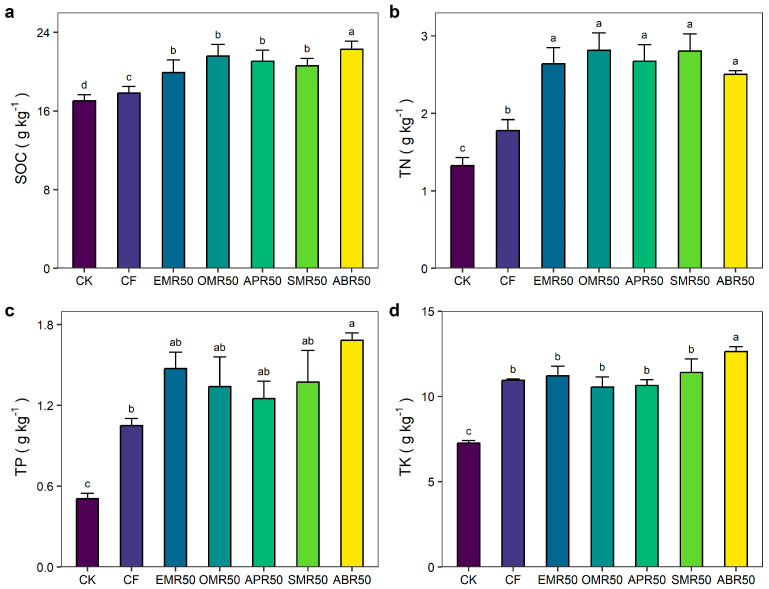
Effect of different waste mushroom residue application on soil fertility. (**a**) Effect of different waste mushroom residue application on SOC. (**b**) Effect of different waste mushroom residue application on SOC. (**c**) Effect of different waste mushroom residue application on TP. (**d**) Effect of different waste mushroom residue application on TK. SOC, TN, TP, and TK indicate soil organic carbon content, total nitrogen, total phosphorus, and total potassium, respectively. CK, CF, EMR50, OMR50, APR50, SMR50, and ABR50 indicate no fertilization, mineral NPK fertilizers, and NPK fertilizer combined with Enoki mushroom residue, Oyster mushroom residue, *Auricularia polytricha* mushroom residue, Shiitake mushroom residue, and *Agaricus bisporus* residue. Different lowercase letters indicate significant differences among treatments (*p* < 0.05).

**Figure 2 plants-13-00663-f002:**
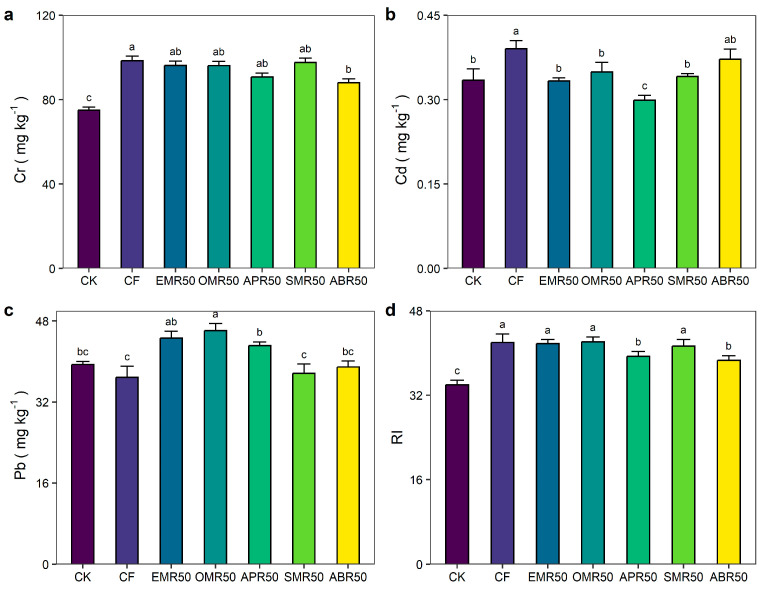
Effect of mushroom residue application on heavy metal contents and the cumulative risk. (**a**) Effect of different waste mushroom residue application on Cr. (**b**) Effect of different waste mushroom residue application on Cd. (**c**) Effect of different waste mushroom residue application on Pb. (**d**) Effect of different waste mushroom residue application on RI index. Cr, Cd, Pb, and RI indicate chromium, cadmium, lead, and potential ecological risk. CK, CF, EMR50, OMR50, APR50 SMR50, and ABR50 indicate no fertilization, mineral NPK fertilizers, and NPK fertilizer combined with Enoki mushroom residue, Oyster mushroom residue, *Auricularia polytricha* mushroom residue, Shiitake mushroom residue, and *Agaricus bisporus* residue. Different lowercase letters indicate significant differences among treatments (*p* < 0.05).

**Figure 3 plants-13-00663-f003:**
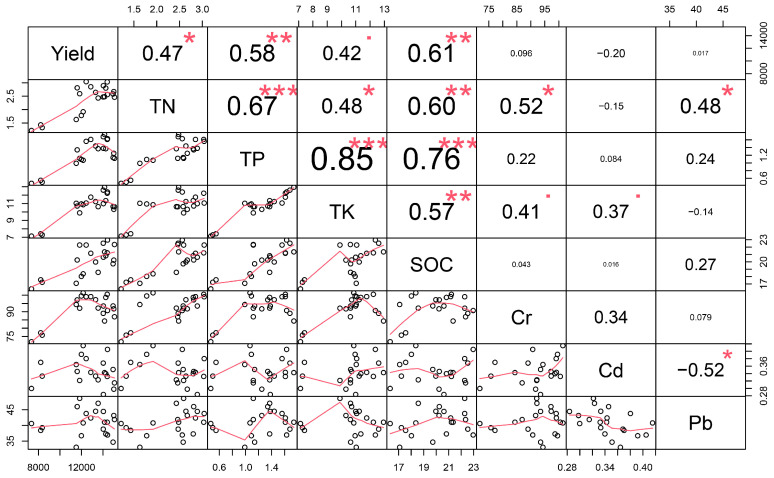
Relationships between soil nutrients and crop yield and heavy metals. The black circles indicate the observed data. The red line indicates the fitted line. * *p* < 0.05, ** *p* < 0.01, *** *p* < 0.001. The red square represents a strong tendency towards statistical significance (*p* = 0.051). Yield, TN, TP, TK, SOC, Cr, Cd, and Pb indicate crop yield, total nitrogen, total phosphorus, total potassium, soil organic carbon, chromium, cadmium, and lead, respectively.

**Figure 4 plants-13-00663-f004:**
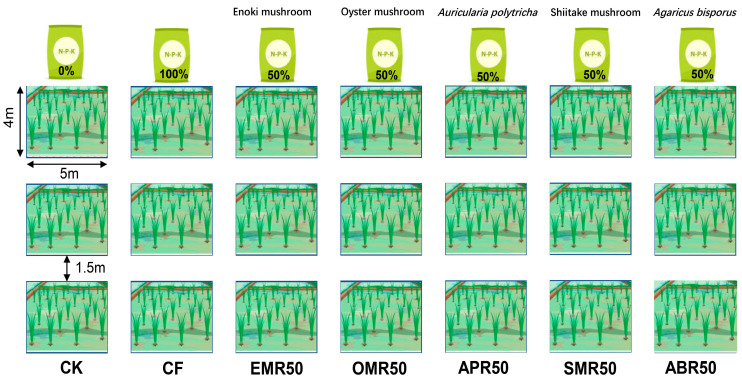
Schematic diagram of field fertilization management.

**Table 1 plants-13-00663-t001:** Effect of different waste mushroom residue on grain yield.

Treatment	Grain Yield (kg ha^−1^)	
	Rice	Wheat
CK	4673.4 ± 557.3 d	3300.8 ± 175.8 d
CF	7740.0 ± 206.1 bc	4148.7 ± 131.0 c
EMR50	8416.1 ± 10.0 b	4827.2 ± 347.3 ab
OMR50	7093.0 ± 62.6 c	4864.5 ± 492.7 ab
APR50	9569.5 ± 338.7 a	5133.0 ± 238.9 a
SMR50	9325.6 ± 408.5 a	5202.0 ± 84.2 a
ABR50	9316.2 ± 300.3 a	4878.3 ± 234.9 ab

CK, CF, EMR50, OMR50, APR50, SMR50, and ABR50 indicate no fertilization, mineral NPK fertilizers, and NPK fertilizer combined with Enoki mushroom residue, Oyster mushroom residue, *Auricularia polytricha* mushroom residue, Shiitake mushroom residue, and *Agaricus bisporus* residue. Values are mean ± standard deviation (*n* = 4). Values followed by different letters in superscript are significantly different among treatments (*p* < 0.05).

**Table 2 plants-13-00663-t002:** The basic information and application ratio of different mushroom residue.

Variable	Oyster Mushroom	*Auricularia polytricha*	Shiitake Mushroom	*Agaricus bisporus* Mushroom	Enoki Mushroom
pH	7.8	5.53	7.4133	7.5233	7.17
C %	35.0	25.3	45.5	15.7	22.2
N %	1.7	1.5	0.8	0.5	1.1
P %	0.3	0.4	0.4	0.4	0.7
K %	0.9	0.5	0.5	1.8	1.8
Pb mg kg^−1^	11.0	1.0	1.6	12.8	94.7
Cd mg kg^−1^	0.2	0.0	0.1	0.1	2.9
Cr mg kg^−1^	10.4	2.3	2.9	34.4	32.7
Input rate (t/ha)	9.7	11.0	19.9	30.8	14.7

## Data Availability

Original data may be provided upon request.

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
