# Peer review of "Responses of Crop Yield, Soil Fertility, and Heavy Metals to Spent Mushroom Residues Application"

_plants, 2024, doi:10.3390/plants13050663_

Round 1
Reviewer 1 Report
Comments and Suggestions for Authors
Dear Authors,
The submitted manuscript titled „Responses of crop yield, soil fertility and heavy metals to spent mushroom residues application: field study and meta-analysis” contains very interesting results, which might interest an international audience. In my opinion the manuscript is generally well-written. Nevertheless, I have found some imperfections, which- in my opinion- should be corrected or at east clarified before an eventual publication. Please, find them below:
1. In my opinion the choice of each mushroom species should be better justified in chapter Introduction.
2. Material and methods. I suggest to add the Figure presenting the experimental design.
3. Tables and Figures should be self-explanatory. Please, add the meaning of abbreviations in the captions.
4. I encourage Authors to enlarge Discussion section and compare the obtained outcomes with greater number of literature sources.
5. Please, look into below listed papers. Perhaps, some of them will be useful in manuscript improvemets:
• Chen et al. 2022. Short-term responses of soil nutrients, heavy metals and microbial community to partial substitution of chemical fertilizer with spent mushroom substrates (SMS). Science of The Total Environment, 844, 157064. https://doi.org/10.1016/j.scitotenv.2022.157064
• Rajawat et al. 2022. An emerging bio-fertilizer for improving soil health and plant productivity. Editor(s): Harikesh Bahadur Singh, Anukool Vaishnav. New and Future Developments in Microbial Biotechnology and Bioengineering, Elsevier, 345-354. https://doi.org/10.1016/B978-0-323-85579-2.00010-1
• Wu, C.-Y.; Liang, C.-H.; Liang, Z.-C. Evaluation of Using Spent Mushroom Sawdust Wastes for Cultivation of Auricularia polytricha. Agronomy 2020, 10, 1892. https://doi.org/10.3390/agronomy10121892
• Rezaeian, S., Pourianfar, H.R. & Attaran Dowom, S. Quantitative Changes in the Biochemical and Mineral Composition of the Substrate in Solid-State Cultivation of Enoki Mushroom. Waste Biomass Valor 12, 4463–4474 (2021). https://doi.org/10.1007/s12649-020-01340-7
Author Response
Comments 1: The submitted manuscript titled „Responses of crop yield, soil fertility and heavy metals to spent mushroom residues application: field study and meta-analysis” contains very interesting results, which might interest an international audience. In my opinion the manuscript is generally well-written. Nevertheless, I have found some imperfections, which- in my opinion- should be corrected or at east clarified before an eventual publication. Please, find them below:
|
Response 1: Thank you for reviewing our paper. We appreciate your comments and suggestions to improve the paper for publications.
|
Comments 2: In my opinion the choice of each mushroom species should be better justified in chapter Introduction. |
Response 2: Thank for your comments. We have added the detail of the choice of each mushroom species in the revised manuscript as followed: “As the important producer of mushrooms in China, Sichuan mainly produces Enoki mushroom, Oyster mushroom, Auricularia polytricha mushroom, Shiitake mushroom, and Agaricus bisporus.”
Comments 3: Material and methods. I suggest to add the Figure presenting the experimental design.
Figure 4 Schematic diagram of field fertilization management.
Comments 4: Tables and Figures should be self-explanatory. Please, add the meaning of abbreviations in the captions.
Response 4: Thanks for your comments. We have added the meaning of abbreviations in the captions in the revised manuscript as followed: “CK, CF, EMR50, OMR50,APR50,SMR50, and ABR50 indicate no fertilization, mineral NPK fertilizers, and NPK fertilizer combined with Enoki mushroom residue, Oyster mushroom residue, Auricularia polytricha mushroom residue, Shiitake mushroom residue, and Agaricus bisporus residue.” “Yield, TN, TP, TK, SOC, Cr, Cd, and Pb indicate crop yield, total nitrogen, total phosphorus, total potassium, soil organic carbon, Chromium, Cadmium, and lead, respectively.”
Comments 5: I encourage Authors to enlarge Discussion section and compare the obtained outcomes with greater number of literature sources.
Response 5: Thanks for your suggestion. We have enlarged the Discussion section and compared the obtained outcomes with greater number of literature sources in the revised manuscript as followed: “For instance, Medina et al. (2012) showed that waste mushroom residue application can enhance soil nutrients and enzyme activities[35]. Kwiatkowski et al. (2021) found that waste mushroom residue application can improve yield quality[36]. Ma et al. (2021) observed that the application of mushroom residue can improve soil nutrient, enhance leaf nutrients, and suppress weeds[37]. Paredes et al. (2016) found that the application of waste mushroom residue can improve soil fertility and crop yield[38]. Chen et al. (2022) reported that substituting of chemical fertilizer with waste mushroom residue can increase soil nutrients, such as SOM, TN, AN, TP, AP, TK and AK[23].”
Comments 6: Please, look into below listed papers. Perhaps, some of them will be useful in manuscript improvemets: • Chen et al. 2022. Short-term responses of soil nutrients, heavy metals and microbial community to partial substitution of chemical fertilizer with spent mushroom substrates (SMS). Science of The Total Environment, 844, 157064. https://doi.org/10.1016/j.scitotenv.2022.157064 • Rajawat et al. 2022. An emerging bio-fertilizer for improving soil health and plant productivity. Editor(s): Harikesh Bahadur Singh, Anukool Vaishnav. New and Future Developments in Microbial Biotechnology and Bioengineering, Elsevier, 345-354. https://doi.org/10.1016/B978-0-323-85579-2.00010-1 • Wu, C.-Y.; Liang, C.-H.; Liang, Z.-C. Evaluation of Using Spent Mushroom Sawdust Wastes for Cultivation of Auricularia polytricha. Agronomy 2020, 10, 1892. https://doi.org/10.3390/agronomy10121892 • Rezaeian, S., Pourianfar, H.R. & Attaran Dowom, S. Quantitative Changes in the Biochemical and Mineral Composition of the Substrate in Solid-State Cultivation of Enoki Mushroom. Waste Biomass Valor 12, 4463–4474 (2021). https://doi.org/10.1007/s12649-020-01340-7
Response 6: Thanks for your suggestion. We have cited some of these references in the revised manuscript. |

Reviewer 2 Report
Comments and Suggestions for Authors
The manuscript titled “Responses of crop yield, soil fertility and heavy metals to spent mushroom residues application”, is an interesting paper. The findings of the study contributes to the advancement of a circular economy in the mushroom sector and are important for the safeguarding the soil environment amid global climate change and food security concerns.
I only have a suggestion for the Materials and Methods:
In “Experimental design”, please include some more details on mushroom residues. Were it applied fresh or dry. How the residues were prepared for application and how was the application done? Also please indicate the application time.
Author Response
Comments 1: The manuscript titled “Responses of crop yield, soil fertility and heavy metals to spent mushroom residues application”, is an interesting paper. The findings of the study contributes to the advancement of a circular economy in the mushroom sector and are important for the safeguarding the soil environment amid global climate change and food security concerns.
|
Response 1: Thank you for reviewing our manuscript. We appreciate your comments and suggestions to improve the paper for publication.
|
Comments 2: I only have a suggestion for the Materials and Methods: In “Experimental design”, please include some more details on mushroom residues. Were it applied fresh or dry. How the residues were prepared for application and how was the application done? Also please indicate the application time. |
Response 2: Thanks for your comments. We have provided the detail of fertilizer management in the revised manuscript as followed: “The waste mushroom residue is sourced from edible mushroom cultivation institutions in the Chengdu Plain. For rice cultivation, 40% of the N fertilizer is applied as basal fertilizer, 40% at tillering stage, and 20% at booting stage. For wheat cultivation, 50% of the N fertilizer is applied as basal fertilizer, 25% at seedling stage, and 25% at jointing stage. P, K fertilizers, and fresh waste mushroom residue are applied annually before plowing as basal fertilizers. All basal fertilizers and mushroom residue are evenly spread on the soil surface by hand and immediately plowed into the plow layer. The plowing depth is 20 cm, first using a plow and then a harrow. Fertilized plots and unfertilized plots are uniformly plowed.” |
